From effective biocontrol agent to successful invader: the harlequin ladybird (Harmonia axyridis) as an example of good ideas that could go wrong

Camacho-Cervantes Morelia mcamacho@iies.unam.mx 1
Ortega-Iturriaga Adrián 2
del-Val Ek 1
1 Instituto de Investigaciones en Ecosistemas y Sustentabilidad, Universidad Nacional Autónoma de México , Mexico
2 Centro de Investigaciones en Geografía Ambiental, Universidad Nacional Autónoma de México , Mexico
Wratten Stephen
Electronic publication date: 2017 May 16
Publication date: 2017
Volume: 5
Electronic Location ID: e3296
Received 2017 Jan 6; Accepted 2017 Apr 11
Copyright: ©2017 Camacho-Cervantes et al.
Copyright year: 2017
Copyright holder: Camacho-Cervantes et al.
License: This is an open access article distributed under the terms of the Creative Commons Attribution License, which permits unrestricted use, distribution, reproduction and adaptation in any medium and for any purpose provided that it is properly attributed. For attribution, the original author(s), title, publication source (PeerJ) and either DOI or URL of the article must be cited.
License URL: https://creativecommons.org/licenses/by/4.0/

Keywords: Invasive threat, Range expansion, Coccinellidae, Biodiversity threats, Perception, Awareness, Invader, Biological control, Introduction vectors, Survey

Funding: The authors received no funding for this work.

==============================
The use of biological control agents to control pests is an alternative to pesticides and a tool to manage invasive alien species. However, biocontrol agents can themselves become invasive species under certain conditions. The harlequin ladybird (Harmonia axyridis) is a native Asian biocontrol agent that has become a successful invader. We reviewed articles containing “Harmonia axyridis” to gather information on its presence and surveyed entomologists researching Coccinellidae around the world to investigate further insights about the current distribution, vectors of introduction, habitat use and threats this species pose. The harlequin ladybird has established populations in at least 59 countries outside its native range. Twenty six percent of the surveyed scientists considered it a potential threat to native Coccinellidae. Published studies and scientists suggest Adalia bipunctata, native to Europe, is under the highest risk of population declines. Strict policies should be incorporated to prevent its arrival to non-invaded areas and to prevent further expansion range. Managing invasive species is a key priority to prevent biodiversity loss and promote ecosystem services.

Introduction

Biological control (or biocontrol) is a pest management strategy whereby natural enemies of a target species are introduced or cultivated to help supress populations (Greathead & Waage, 1983). The target pests are often those that have important negative impacts on crop production or human health (Greathead & Waage, 1983). Biocontrol agents have commonly been used in the last decades to control pests, diseases and weeds as an alternative to chemical pesticides (De Clercq, Mason & Babendreier, 2011). It is considered a more environmentally-friendly way to deal with pests, however, as it can be an efficient strategy to control pests it can also be a route by which potentially damaging alien species are introduced and spread worldwide (Babendreier, 2007).

A famous example of biocontrol that went wrong is that of the Indian mongoose (Herpestes auropuncatus) which was intentionally introduced during the late 19th century and early 20th into over 60 islands mainly to control rats. The Indian mongoose is now a great contributor to the decline of native birds, mammals and herpetofauna in all these islands (Hays & Conant, 2007; Barun et al., 2011). Similarly, the guppy (Poecilia reticulata) was introduced worldwide to control malaria by preying on mosquitoes’ eggs and has now established in over 60 countries across the globe (Deacon, Ramnarine & Magurran, 2011). In Mexico, it is known for threatening native topminnow fish species and getting advantage from associating with them (Magurran, 2009; Valero, Macías Garcia & Magurran, 2008; Camacho-Cervantes et al., 2014). Nevertheless, if selected appropriately, biocontrol agents can provide a useful service without negative side effects. Pearson & Callaway (2005) suggest selecting biocontrol agents that are as specific and efficacious as possible to prevent indirect non-target effects. They point out that agents that are effective at reducing their target species will reduce their own populations through density-dependency.

The harlequin ladybird (Harmonia axyridis, Pallas 1773) is native to Asia, and its distribution ranges from Kazakhstan in the West to the Pacific Coast in the East, including Kyrgyzstan, and from southern Siberia in the north to southern China in the south (Orlova-Bienkowskaja, Ukrainsky & Brown, 2015). However, populations have become established in many countries outside its native range due to its introduction as biocontrol agent and subsequent range expansion (Brown et al., 2011b; Roy et al., 2016). The harlequin ladybird is a generalist species widely used as biocontrol agent mainly due to its voraciousness (Pervez & Omkar, 2006). Harlequin ladybirds have been used as biocontrol agents of aphids and coccids since the 1910’s (Gordon, 1985).

As an invasive species, the harlequin ladybird is considered to be noxious in crop fields as well as to humans (Orlova-Bienkowskaja, 2014; Koch, Venette & Hutchison, 2006; Brown et al., 2011b). They are now considered a successful invasive species due to its polyphagy, wide host range and ability to survive under scarcity of prey to feed on (Pervez & Omkar, 2006). When preys are scarce they recur to prey on other ladybird larvae or cannibalism to survive (Katsanis et al., 2013) and they can also feed on fruit from crop fields (Koch, 2003). In vineyards it was reported that they can taint the wine flavour as individuals get into the wine making process (Koch, 2003). In urban areas it is considered a household pest when it shelters during winter inside houses and buildings (Koch, Venette & Hutchison, 2006). In addition, its bites to humans are compared to those of mosquitoes; however, these are not very common (Ramsey & Losey, 2012).

Studies suggest that the harlequin ladybird may be contributing to displace native Coccinellidae in those areas where it has established (Roy et al., 2012; Yasuda & Ohnuma, 1999; Snyder, Clevenger & Eigenbrode, 2004; Brown et al., 2011a; Bahlai et al., 2015; Honek et al., 2016). For example, it has been found that they can carry spores of a parasitic microsporidia that, while not harming them, are lethal pathogens for other species (Vilcinskas et al., 2013). As other invasive species, the harlequin ladybird has the potential to homogenize ladybird diversity where it establishes populations and with this they might be threatening ecosystem services that can be provided only by an assemblage of species (Cardinale et al., 2003). In this study we aim to document the current non-native distribution of the harlequin ladybird, the origins of the introductions that have led to its current distribution and threats entomologists perceive it is posing on native ladybirds. We reviewed published records and surveyed entomologists working with Coccinellidae worldwide to produce a distribution map and gather insights from the academic community working on the subject.

Methods

Published records review

In order to assess the multiple locations where the harlequin ladybird has been seen or collected, building on Brown et al. (2011b), we used the search engine Web of Science ™ to find published articles from January 2009 to June 2015 that contained the word “Harmonia axyridis”. We focused only in articles that were published in scientific indexed journals that had gone through peer review. We reviewed all the articles found (over 600) to see how many contained information on the presence of the harlequin ladybird in a particular place or region. From all the articles reporting harlequin ladybird presence we built a database (attached as Supplemental Information) that contains the site where this species was seen or collected, the year of sight (when absent year of publication of the article was included with an asterisk), the geographic coordinates of the site, and name and affiliation of the first author of the article.

Survey

To complement our literature review, we designed a survey that allowed us to document entomologists’ non-published records of presence of the harlequin ladybird and their ideas on its invasion routes and threats to local species. We acknowledge the results from our survey might not be as strict as published records, nevertheless this information is useful for future policy design, conservation plans and environmental education programs (Martin et al., 2012). In order to mitigate the effects of psychosocial and motivational bias in our survey, we followed Sutherland & Burgman (2015) recommendations to design our survey and select the specialists we were to contact: we contacted experts independently, gave the same importance to all responses (not relying only in the best-regarded expert), ensured our questions were specific and unambiguous, and contacted as many experts as we could. All our queries were open questions so we could document as many details as possible. It was essential for us to give the respondents the opportunity to detail their views in their own words. In this sense, this section of our study can be considered within a qualitative approach (Maxwell, 2013). Our survey was designed to address the following questions (see exact survey template in the Supplemental Information): (1) Are you aware of harlequin ladybirds within your surroundings or where you carry out your research; (2) If yes, are they throughout the region or in localised parts only; (3) Do you know anything about the origin of their introduction; (4) Are you aware of any negative effects the harlequin ladybird might be causing; (5) Are you aware of any native species gone extinct due to the presence of the harlequin ladybird? Aside from these questions we gave respondents the option to add any other information they thought could be relevant to us.

The survey was intended for Coccinellidae specialists. To obtain a list of researchers studying Coccinellidae, using the Web of Science ™ engine, we searched for all published articles from December 2000 to April 2015 that contained the word “coccinellidae”. We collected the name, affiliation and e-mail of corresponding authors of all the articles we could. With this we ensured to contact scientist that are familiar with ladybirds’ species. In total we were able to gather the contact information of 473 scientists from 65 different countries. We sent an email to all of them including the survey and our goals. Emailing the survey was the fastest way to contact scientist while allowing them to dedicate as much time as they needed to respond our survey and attach any further information they felt was relevant. We organised all the responses we received to construct summary figures and tables that include all opinions expressed.

Maps

We used Google Earth software to create the databases containing the different locations mentioned both in the literature review and the survey to create the maps. The files generated were then transferred to ArcGIS® for processing, layer overlapping and layout design. Both maps show in darker colour the native distribution reported by Brown et al. (2011b) and Koch, Venette & Hutchison (2006). Reports found in the literature review and through the survey are all referred to administrative units, ranging from locality to country, thus some countries are entirely coloured. In these cases, harlequin ladybirds do not necessarily occur throughout the country. In the survey scientists reported harlequin ladybird’s presence at a local, province or state and region scale; in order to keep valuable information provided by the respondents, different symbols representing different geographical scale were used. Given that some reports referred to countries and the whole country had to be coloured, invasion reported for bigger countries might seem worse than invasion reported for smaller ones. However, impacts on ecosystems and human livelihoods are of local attention no matter the country size.

Results

Published records review

From the over 600 articles that contained the key words “Harmonia axyridis” reviewed, in total we found 153 that actually reported information on the presence of the harlequin ladybird. Given that we set Brown et al. (2011b) as a starting point for our review we focused on articles published from 2009 to 2015. However, we included 34 extra articles that provided information on the presence of the harlequin ladybird from earlier dates (1988–2008) that were not included in Brown et al. (2011b) revision. The harlequin ladybird was reported present or collected in published articles in 339 sites around the world, in 58 countries outside its native range (seven in Africa, 12 in America, two in Asia, 36 in Europe, and one in Oceania; including reports from Brown et al. (2011b) and Roy et al. (2016), Fig. 1).

Figure 1 Published records of the presence of the harlequin ladybird in the world.

Survey

From the 473 surveys we sent, we received back 74 filled (15.6%) from 35 different countries (one in Africa, 10 in America, five in Asia, 17 in Europe and two in Oceania). Additionally we included reports that were published in grey literature (e.g., bulletins, science communication webpages, citizen science databases, news, etc.) as they had not gone through the peer review process, and we were able to identify the presence of the ladybird in eight more locations through reports from colleagues that included a picture.

The harlequin ladybird was reported present in 28 countries, but not in Australia, Cuba or the Philippines (Fig. 2). Certain areas of Brazil, Chile, Czech Republic, India, Iran, New Zealand, Portugal, Turkey and the USA were mentioned to be free of the harlequin ladybird, but there are presence reports in other areas of these countries (Fig. 2). Harlequin ladybirds were reported to inhabit urban areas, crop fields, greenhouses, pasture fields, meadows, forests and natural reserves (Fig. 3). From the 74 surveyed scientists, 19 (25.7%) mentioned the harlequin ladybird might be posing a threat to native Coccinelidae and other arthropods diversity. Adalia bipunctata was mentioned by five experts as the most likely to present population declines after the harlequin ladybird invasion, and a researcher from Venezuela mentioned Cycloneda sanguinea, Hippodamia convergens and Colleomegilla maculata to be under threat. As for the vectors of introduction, range expansion and biocontrol were the most mentioned (Fig. 4). Some scientists reported that the harlequin ladybird arrived to their countries accidentally by human related activities (transportation and trade of goods). This ladybird is considered a nuisance in China, the USA, Brazil, and the Czech Republic. In the USA and Venezuela, it was reported to compete for resources with native species.

Figure 2 World distribution of the harlequin ladybird from survey records.

Presence is represented by filled circles and absence is represented by empty squares.

Figure 3 Habitat types entomologists report the harlequin ladybird uses by countries.

∗Countries where the harlequin ladybird is native. + The harlequin ladybird is known to be established in other than urbanised areas in Japan, though this was the only habitat mentioned by an expert in the survey.

Figure 4 Reported vectors of harlequin ladybird’s introduction.

Discussion

Combining our literature review and our survey, we found the harlequin ladybird is reported in 59 countries outside its native range, 11 more countries (Ecuador, Egypt, India, Lithuania, Moldova, New Zealand, Pakistan, Swaziland, Tanzania, Turkey and Venezuela) since Brown et al. (2011b) review. Presence of the harlequin ladybird in Pakistan was reported from our survey only. Similarly, presence of the harlequin ladybird in Auckland, New Zealand, was just recently (May 2016) reported by the Ministry of Primary Industries of the New Zealand Government. To the best of our knowledge, there are no published records in scientific journals from these countries yet.

We consider a survey is a fast and useful way of gathering information that could otherwise require significantly more time and effort; however, information drawn from our survey must be interpreted with caution because data has not been published and therefore has not gone through peer review process. In order to increase the soundness of our survey, we contacted scientists that have published studies about Coccinellidae and are familiar with ladybird species. The information drawn from our survey provides insights that could point out priority areas of research, for example species potentially under higher risk of population declines or countries that could be under higher risk of invasion by expansion range. In the following sections, we will discuss the information we were able to obtain from our survey looking at each continent separately.

Asia

The harlequin ladybird is native mainly to the East part of this continent, and it was considered invasive in the Southern part of Kazakhstan and Kyrgyzstan; it was not found there during the 19th century, and in the late 1960s it was introduced as a biological control agent (Roy et al., 2016). However, in 2015, Orlova-Bienkowskaja et al. published a revised native distribution of the species and reported harlequin ladybirds could be expanding its native range to Kazakhstan and Kyrgyzstan using the Turkestan-Siberian Railway. These populations are considered to be related with natives in Asia, as they are morphologically and genetically similar to them (Orlova-Bienkowskaja, Ukrainsky & Brown, 2015). Because of the latter, in our distribution maps we decided to consider Kyrgyztan as part of the native range.

One scientist from Japan and one from China related to us that in their countries the harlequin ladybird is considered beneficial and a useful biocontrol agent. In China, he said, scientists are still working on developing effective ways of mass rearing them for biocontrol purposes. However, another scientist from China recounted to us that some people do perceive this ladybird as a nuisance.

Oceania

Brown et al. (2011b) and Roy et al. (2016) found there were no reports of the harlequin ladybird being present in Oceania. We found in a bulletin from the Ministry of Primary Industries of the New Zealand Government the ladybird was seen in Auckland, this does not mean it is established there but it is the first time it is reported alive in the wild. According to a respondent from our survey and Brown et al. (2011b) the harlequin ladybird has been intercepted a couple of times in Australia but individuals were either dead or caught before entering the wild.

In Oceania, biological control plays an important role in natural resource management, thus there is a continuous scientific effort to improve both its understanding and application (Julien et al., 2007). The reason behind the interceptions of the harlequin ladybird is that Australia has implemented advanced detection, prevention and impact mitigation programmes that include species distribution models and pre-border risk assessments before importing species (Pheloung, Williams & Halloy, 1999).

America

First introduction of the harlequin ladybird in North America was in the USA in 1916, but the species was not considered established until 1988 (Brown et al., 2011b). The harlequin ladybird had a rapid period of invasive expansion range after the introduction in eastern North America, which acted as source of colonists for Europe, South America and Africa (Lombaert et al., 2010). A reduced dependence on photoperiod to trigger reproduction was thought to contribute to the development of this invasive population (Reznik et al., 2015).

Despite the well studied nuisance that the harlequin ladybird represents, in North America it was believed to be more efficient than other native Coccinellidae to control pests and therefore it was encouraged to be used as a biocontrol agent (Lucas, Gagne & Coderre, 2002). Our results show the harlequin ladybird is present almost continuously from North to South in America. It has not been reported only in Central America (from Guatemala to Panamá), Guyana, Surinam and Bolivia.

A scientist from Brazil told us in his response to our survey that biocontrol companies in his country are still rearing harlequin ladybirds and selling them; according to him, the harlequin ladybirds are competing with native aphidophagous species but there are not yet records of a species going extinct after the harlequin ladybird was established. Similarly, an expert from Chile said scientists in the country are concerned about the negative effects of the harlequin ladybird’s invasion as it is now the most abundant ladybird in Chile, but there are still people who like them and are keen to promote their permanence in the country. An expert from Chile and an expert from the USA told us that harlequin ladybirds in their countries have been seen in the company of native and other introduced ladybird species, meaning that they utilise similar resources in the same areas. A researcher from Venezuela related to us that the harlequin ladybird is believed to be competing with native species for resources in his country. According to two researchers from the USA, the harlequin ladybird is considered a nuisance to humans because they overwinter in buildings, especially in the northern part of the country, where winters in the wild can be too harsh for them.

Europe

Introductions in Europe started in the East, in Ukraine and Belarus, during the 1960s and in the West in France in 1982 (Brown et al., 2011b). Brown et al. (2011b) predicted in his review that invasive populations of the harlequin ladybirds from Eastern Europe would meet invasive populations of the West. Our distribution maps (Figs. 1 and 2) show that after six years this prediction is most likely true. Moldova, Lithuania and Turkey have now reported populations and from the remaining 12 countries in Europe without reports, Andorra, Monaco, San Marino and the Vatican are surrounded by countries already invaded. Cyprus, Iceland and Malta do not have reports of invasion; being islands, their ecosystems are more vulnerable to invasion and thus further research must be carried out to evaluate if the harlequin ladybird is present or not.

Each country, and in bigger countries each state is autonomous to permit or not the use of certain biocontrol agents; lack of or deficient regulations can affect neighbouring entities and end up in species being imported to an area and then “naturally” spread where they were not permitted. In Switzerland an expert indicated us that harlequin ladybird’s use as biocontrol agent is not permitted; nevertheless, he said it is now the most abundant Coccinellidae species in that country. Indeed, in Switzerland the harlequin ladybird was reported present for the first time in 2006, and it was considered abundant at the time (Eschen et al., 2007). In Ireland, a researcher told us the harlequin ladybird arrived in celery hearts shipped to a supermarket, but he believes the establishment of this ladybird is delimited by climate conditions. A Swedish scientist said, that given the current climate warming trends, half of Sweden could be suitable habitat for the harlequin ladybird. In the Netherlands, the harlequin ladybird was referred by a scientist to be now attacked by natural enemies in the country. In France, they have been seen overwintering with native species, and in the Czech Republic a researcher explained that harlequin ladybirds are competing with native species for food resources. According to our survey in England, Switzerland and the Czech Republic, Adalia bipunctata is the species under the highest risk of population declines, which is consistent with results of previous studies that show native ladybird populations in Europe are declining after the harlequin ladybird introduction and establishment (Roy et al., 2012; Brown et al., 2011a).

Africa

The harlequin ladybird was reported present for the first time in Tunisia, Egypt and in South Africa in the early 2000s (Brown et al., 2011b). Limited data from Africa did not allow us to see if this species is present along the latitudinal gradient from Egypt to South Africa, but it is now reported in Tanzania and Swaziland too. A researcher from South Africa told us that the harlequin ladybird feeds on the senescent oak leaves’ aphids. We believe that the information gap in most of Africa (both in the literature review and the survey) is due to the lack of research done for the species in the continent.

The most cost-effective strategy to diminish biodiversity loss due to invasive species is to prevent them arriving rather than mitigate their effects afterwards (Mack et al., 2000). Identifying vectors of species introduction outside its native range is a key measure to avoid invasion and improve management strategies (Puth & Post, 2005; Hulme et al., 2008). Dominant vectors of introduction for the harlequin ladybird according to our survey are biocontrol and expansion range, the latter being more difficult to address. Given that the harlequin ladybird has been found in 59 countries outside its native range (as south as Punta Arenas in Chile and as north as Oslo in Norway), we are inclined to think it is capable of surviving in all type of habitats and tolerate a wide climate range (Roy et al., 2016). We suggest the harlequin ladybird is present in many more countries than what we currently know, including central Africa.

Conclusions

At the country level new appearances are expected to decrease, since the current distribution has already reached a wide global range. However, more research is needed at regional and local scales in order to add more detailed information to the current knowledge. Citizen science—engaging the public in a scientific project (Bonney et al., 2014)—has invaluably contributed to the monitoring of invasive species while engaging the public in a hands-on way with ecological problems and its potential solutions. Getting the general public involved in the gathering of scientific data is of great use to address spatial questions, such as the distribution of an invasive species (Kobori et al., 2016). Along with the distribution maps produced using published records in scientific articles, like ours, there are increasing efforts to engage the public in the monitoring of the harlequin ladybird to be more accurate when describing its expansion. Indeed, in Chile (Chinita Arlequín), France (L’Observatoire de la Coccinelle asiatique en France), United Kingdom (The Harlequin Ladybird Survey) and the United States of America (The Lost Ladybug Project) there is a continuous monitoring of the harlequin ladybird dispersion with the help of citizens.

Classic biological control does not necessarily have negative outcomes; it can promote biodiversity conservation as well as being a greener alternative to chemical treatment for pests (Davies & Britton, 2015). However, it should be treated carefully; to prevent large effects on ecosystems, biocontrol species should be at least specialists and its individuals free of parasites. Biological control strategies’ outcomes can be quite complicated to predict, especially when agents used are exotic to the area where they are to be used (Louda & Stiling, 2004). To predict ecological consequences data must be collected outside quarantine areas, where biological control agents are usually kept during assessments, and ecological risk criteria must be acknowledged before selecting targets; some pest species might just be unsuitable to be controlled using a biocontrol agent because they have many native relatives that are not pests and would be at risk of being suppressed (Louda et al., 2003). When research on life history, dispersal, phylogeny and behavioural ecology of species to be used as biocontrol is considered there is an increase in the accuracy of side effect predictions (Messing & Wright, 2006). Managers must always consider effects that altering the food web structure may have on ecosystems, as the invasive species population could be contained or eradicated but native species could get harmed (GonzáLez-Chang et al., 2016; Louda et al., 2003).

Interdisciplinary collaboration is needed for a better management of invasive species mobility to analyse and potentially predict introduction and spread of invasive species (Banks et al., 2015). Identifying areas where policies could benefit from synergies between climate predictions, land use change and invasive species management can help prevent future invasion (Bellard et al., 2013). Assessments that consider the risk of introducing a classical biological control agent and the risk of not doing it must be carried out for every case; however, we must acknowledge biocontrol agents will always present a certain environmental risk (De Lange & Van Wilgen, 2010).

Supplemental Information

Supplemental Information 1 Published records of Harmonia axyridis seen or collected

Click here for additional data file.

Supplemental Information 2 Survey template

Click here for additional data file.

Supplemental Information 3 Database of survey responses

Click here for additional data file.

We are truly grateful to all the scientists who agreed to answer our survey and share with us valuable information and perceptions (full list as Supplemental Information 3). We thank Daniela Arellano for organising the records of presence of the harlequin ladybird and contacts’ information into a comprehensive database. We thank Sara P Mason for helping with the online distribution of the survey and two anonymous reviewers for constructive comments.

Additional Information and Declarations

Competing Interests

Author Contributions

Data Availability

The authors declare there are no competing interests.

Morelia Camacho-Cervantes conceived and designed the experiments, performed the experiments, analyzed the data, contributed reagents/materials/analysis tools, wrote the paper, prepared figures and/or tables, reviewed drafts of the paper.

Adrián Ortega-Iturriaga performed the experiments, analyzed the data, contributed reagents/materials/analysis tools, wrote the paper, prepared figures and/or tables, reviewed drafts of the paper.

Ek del-Val conceived and designed the experiments, analyzed the data, contributed reagents/materials/analysis tools, wrote the paper, prepared figures and/or tables, reviewed drafts of the paper.

The following information was supplied regarding data availability:

The raw data has been supplied as a Supplementary File.

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
