# Peer review of "From effective biocontrol agent to successful invader: the harlequin ladybird (Harmonia axyridis) as an example of good ideas that could go wrong"

_PeerJ, doi:10.7717/peerj.3296_

## Round 0.1 · original submission · Minor Revisions

· Academic Editor

Minor Revisions

Dear authors

I appreciate that you have been buffeted a bit by past referees. Now, the 2 referees that I chose have worked hard on your ms and do require changes

Whether you consider them as major or minor is up to you-but they need to be dons as best you can please

·

Basic reporting

There are a number of places where the English could be improved. I have not corrected every mistake - see General comments below for some examples - but I also recommend that you get a native English speaker to read through your manuscript before resubmitting. I also think that the structure of the discussion is quite poor. It rather hops from one topic to the next at the moment. Consider why people might refer to your paper - I would have thought that on this global scale, readers might seek your paper to understand what is happening in a certain region of the world. A more appropriate structure therefore, might be to look at each region in its own section of the discussion, and to consider the risk facing the as yet unaffected neighbouring countries.

In terms of figures, while you refer to figures 3 and 4 in the results section, there did not seem to be much reference to the findings in the discussion. If these results are not that interesting, they should not be included. Figure 4 is not a very good figure because the information could more easily be understood if the raw data were shown - a good graphic should simplify data, not make it more difficult to comprehend. I also think that figure 5 would be better represented as a table, with countries listed and ticks added for the ways the ladybird has entered the country.

I question the usefulness of Table 1. This is not the kind of table a reader is drawn to looking at. This information could be in the main text.

While the manuscript is "self-contained" and has results that match up to the research question, I think that some more reference has to be made to the Roy et al paper in the discussion. You and they have produced similar types of map, but how does yours differ? Does it add any new information? It appears to, so why not make more of that?

Experimental design

I think your research question is slightly vague and broad and could be improved. For example, you do not go into detail about the insights you seek from entomologists, nor what these insights might help with. In addition, your rationale seems to be to raise awareness of the importance of selecting biocontrol agents carefully. However, surely this awareness raising has been going on for at least two decades? See any text book on biological control, or papers such as the two listed below, or any by authors that have done much in the area such as Mark Hoddle, and you will find guidelines for selecting appropriate biological control agents, because we now know that selecting the wrong ones has deleterious effects. I think therefore, that this should not be you main reason for this study. Also on this point, you need to emphasise WHEN the harlequin ladybird was introduced to the different countries. I would be surprised if it was a recent biological control agent introduced to a new country, given the lessons we have learnt countless times over. This important context is missing throughout the paper.

Refs:
Julien M.H., J.K. Scott, W. Orapa, and Q. Paynter. 2007. History, opportunities and challenges for biological control in Australia, New Zealand and the Pacific Islands. Crop Protection 26: 255-265.
Messing R.H. and M.G. Wright. 2006. Biological Control of invasive species: solution or pollution? Frontiers in Ecology and Environment 4: 132-140

I also have a problem with how you have reported some of your results. In the discussion, you make statements based on the responses to your survey, but ideally these statements should be corroborated with other evidence. How can we trust the statements? Are they made by lab technicians or professors? Are the backed up or refuted by other evidence? Where you are going to include some statement from your surveys, like "the Harlequin ladybird is the most dominant ladybird in Switzerland", you need to either back it up with facts, or state clearly that it is someone's opinion (preferably a named person).

Validity of the findings

I think the findings are valid - you have found some countries where the ladybird is present where Roy et al. said it was absent, and you have spoken to some people about their take on the matter, highlighting how unaffected countries might be need stricter controls to prevent invasion. However, I think the way it is framed could be better. As mentioned above, if you could include and emphasise the timing aspect and cross-reference that with the mechanism by which these things have invaded, that would provide some more context. Then if you could turn the discussion round to make it more about highlighting which unaffected countries might be most at risk and how and what they should do about it, that would make a more interesting read, if not a more informative one.

Comments for the author

Overall, it seems like this qualitative approach has taken a lot of work and has been a useful exercise in updating a distribution map without actually doing any of the expensive fieldwork, but I think that to turn it into a useful story, the manuscript needs a better angle. It needs a stronger rationale and research question in the introduction and the discussion needs to be more useful to practitioners. Academics requiring greater depth can, and will, use the Roy et al. paper, but to make this contribution useful, I think you should use either the alternative angle above or another angle that seems more fitting to you.

Specific comments below:
Abstract:
L23 - biocontrol agents do not eradicate pests, they regulate them.
L24-25 is poorly worded - try something like "However, the biocontrol agents can themselves become invasive species under certain conditions"
L30 instead of saying 19 scientists, insert the percentage here. The reader has no way of knowing if 19 is a high or low number

Introduction
L44-46 is poorly written. What about: "Biological control is a pest management strategy whereby the natural enemies of a target species are introduced or cultivated to help suppress populations. The target pests are often those that have important negative impacts on crop production, human health or biodiversity..."
L49 - nevertheless is the wrong word. I think you mean "however". In any case, this sentence also needs reworking.
L53 - when did the Indian mongoose example occur or become known?
L56 "Alike" is not the right word here. "Simarlarly"?
L70 & L71 insert the word "agent" after biocontrol
L75 No "s" on the end of prey
L79 - make a new sentence at "in vineyards..."
L82 - use "although" instead of "however"
L88 and elsewhere - change "were" to "where" - get an English speaker to check if you are not sure where to change this

Methods
Did you check the grey literature as well as papers? If not why not?
L118 - more detail is needed on the Sutherland and Burgman reccommendations
L123 - sometimes Harlequin has a captial H and sometimes a small h. Choose one and be consistent
L123 - 128 - are these the actual questions you asked? If not, can you include them in the Supp Mat? Also, was any information recorded about the level of scientist the respondent was? Lab Tech, Post doc, Professor etc? Could that have impacted the results or the validity of their answers?

I think the discussion should include some kind of critique of your methods. Was your sampling method suitable? Were you questions approrpriate? How could it have been done better?

L141 A lot of the information here should be in the caption for the map or self explanatory from the map legend. Besides, the map should be in the results section.
L153 -155 - this does not make sense to me. What do you mean by appreciated?

Results
L159 More details are needed for this section. How many articles did you find? What years were they from? What countries, journals etc were they from? How did they contribute to the map? Have a look at other metanalysis type papers to see what to present here.

Were there any conflicting reports from the scientists responses or did everyone agree?

Discussion
L204 -206 - you need to explain this a bit more
L227 Here is an example of where your reporting of results gets a bit messy. You state here that the Harlequin ladybird is the most abundant species of Coccinellid in Switzerland. Normally, this would be backed up with evidence like a cited paper. You cite Table 1 which is a list of responses from unknown scientists. How can we be sure of this "fact"? Can you back it up with references? Can you cite the person's name as a personal communication? If not, you need to be very clear that this is opinion and not scientific fact.

You could also make reference to any guidlines now in place for selecting and introducing a biocontrol agent, for dealing with the Harlequin ladybird once it has invaded, and/or for preventing it entering/invading.

Reviewer 2 ·

Basic reporting

In their review, Camacho-Cervantes et al., presented an overview of the current Harmonia axyridis worldwide distribution. By reviewing 600 scientific articles found using ISI Web of Science search engine, and by surveys’ results answered by 74 scientists working on Coccinellidae around the world, Camacho-Cervantes et al., were able to achieve their first objective (to document the non-native range of Harlequin ladybird). The current review also highlights the importance of understanding the potential expansion of the introduced biocontrol agents’ distributional range, which might impact native communities even if local policies don’t allow the introduction of exotic species. However, potential biases might appear in trying to quantify this change in distributional ranges by only using surveys. Although the authors acknowledge the limitations of such an approach between lines 114 and 117, the discussion section should also address it; therefore, the conclusions about the origin of current distributional ranges are not too bold.

Experimental design

No experimental design needed. This review has sought their data using ISI Web of Science search engine. No statistics were applied or needed in order to answer the questions stated at the end of the introduction.

Validity of the findings

Overall, the article is well written, with minor grammatical suggestions included as “general comments to the authors”. I recommend the publication of this article, as the results it presents contribute to enhancing our awareness about releasing biocontrol agents outside their native distributional ranges, without an understanding of the impacts of such a strategy on native food webs. Recently, González-Chang et al., (2016) pointed out some issues related to this approach regarding perceiving arthropod interactions as a “food chain”, instead of being interacting in a “food web”. However, despite the strengths of their review, the authors need to clarify the concept of “Biocontrol” used in their manuscript, which in my understanding is referring to “Biological Control”. The authors mentioned several times to the non-target effects and environmental risks of a “Biocontrol” approach. Louda et al., (2003), compiled several cases where a “Classical Biological Control” approach was used to control weeds and arthropods, generating non-target effects on local biodiversity. Conversely, “Conservation Biological Control” can also be understood as a strategy that promotes “Biocontrol”, and not necessarily poses a threat to local biodiversity or to increasing environmental risks. When managed accordingly, conservation biological control strategies can enhance biodiversity (Gurr et al., 2017), and thus promoting several ecosystem services (Wratten et al., 2012). Therefore, a clarification needs to be added to the current manuscript, as different biological control strategies exist.

Comments for the author

Suggested literature
González-Chang, M., Wratten, S. D., Lefort, M.-C. and Boyer, S. (2016) ‘Food webs and biological control. A review of molecular tools used to reveal trophic interactions in agricultural systems.’, Food Webs, 9, pp. 4–11.
Gurr, G., Wratten, S. D., Landis, D. A. and You, M. (2017) ‘Habitat management to suppress pest populations: Progress and prospects’, Annual Review of Entomology, 62, pp. 91–109.
Louda, S. M., Pemberton, R. W., Johnson, M. T. and Follett, P. A. (2003) ‘Nontarget effects-the Achilles’ heel of biological control? Retrospective analyses to reduce risk associated with biocontrol introductions.’, Annual Review of Entomology, 48, pp. 365–396. doi: 10.1146/annurev.ento.48.060402.102800.
Wratten, S. D., Gillespie, M., Decourtye, A., Mader, E. and Desneux, N. (2012) ‘Pollinator habitat enhancement: Benefits to other ecosystem services’, Agriculture, Ecosystems & Environment, 159, pp. 112–122. doi: 10.1016/j.agee.2012.06.020.

Suggested changes in the main text
line 48. environmentally-friendly
Line 79. the crop field (add citation). In vineyards....
Line 175. arthropods diversity. Adalia....
Line 180-181. Awkward wording...and in...and in...Brazil, Czech....
Line 181. Czech. In the Usa and Venezuela...
line 186. delete PRESENT
Line 214. Europe. Limited data....
Line 214. DID NOT instead of DOES NOT?
Line 221. change PREDICT for SUGGEST??
Line 224. change TO BE USED for ITS USE
Line 225. IN Switzerland
Line 226. delete THERE
Line 229. U.S.A. be concise throughout the text (USA used previously)
Line 248. get harmed (add citation).
Line 251. As a result....
Line 255. delete HAS BEEN OBSERVED
Line 281. Arlequín
Line 290. Awkward wording...please rephrase. Also, authors seem to be referring to classical biological control. Please clarify.
Line 292-293. Please clarify to which biological control strategy you are talking about.

---

## Round 0.2 · accepted · Accept

· Academic Editor

Accept

Morelia

Thanks for your good work on this and congratulations on its final acceptance